# Dielectric Behavior of Stretchable Silicone Rubber–Barium Titanate Composites

**Argyri Drymiskianaki** [1,2], **Klytaimnistra Katsara** [2,3], **Alexandra Manousaki** [2], **Zacharias Viskadourakis** [2,*] and **George Kenanakis** [2,*]

1    Department of Materials Science and Technology, University of Crete, GR-70013 Heraklion, Crete, Greece; adrym@materials.uoc.gr
2    Institute of Electronic Structure and Laser (IESL)—Foundation for Research and Technology—Hellas (FORTH), 100 N. Plastira, Vassilika Vouton, GR-70013 Heraklion, Crete, Greece; klyto.katsara@iesl.forth.gr (K.K.); manousa@iesl.forth.gr (A.M.)
3    Department of Agriculture, Hellenic Mediterranean University, Estavromenos, GR-71410 Heraklion, Crete, Greece
*    Correspondence: zach@iesl.forth.gr (Z.V.); gkenanak@iesl.forth.gr (G.K.)

**Abstract:** In this study, elastomer composites, including silicone rubber and barium titanate, were fabricated by mechanical mixing, a low-cost, fast, and easy technique to produce highly dielectric materials. The resulting composites were investigated in terms of their dielectric and mechanical properties in terms of filler percentage in the mixture. Dielectric permittivity measurements were taken using the microwave regime, and uniaxial tensile tests were carried out for the study of the materials' mechanical properties, while combined experiments were also carried out to investigate potential correlations between them. The experimental results show that barium titanate inclusions in silicone matrix significantly improve the dielectric constant while reducing the mechanical properties of composites. In addition, combined experiments show that the composites exhibit a nearly stable dielectric profile under mechanical deformations. Consequently, mechanically mixed barium titanate–silicone elastomer composites could potentially become a cost-effective alternative in the extensive market for insulating materials and flexible electronics.

**Keywords:** dielectric materials; barium titanate; silicone; dielectric permittivity; insulating composites; flexible electronics



## 1. Introduction

Today, rapid technological progress requires constant development of innovative dielectric materials that are not only cost-effective but also adaptable to different applications. The study of dielectric properties is an increasingly interesting topic, as it is intrinsically related to a wide range of technological applications in the field of photonics and electrical insulating materials as well as active dielectric materials [1–4]. To date, the fabrication of high dielectric permittivity materials has been the focus of numerous studies, such as the emerging field of flexible electronics, which is gaining ever-increasing approval in industrial operations, as well as in the health sector, for use in flexible sensors for wearable applications [5,6]. As expected, the goal of functionality in wide-ranging motion requires flexibility in the component materials [7–9]. However, this requirement makes conventional capacitive sensors, composed mainly of rigid silicon, not appropriate for inclusion in flexible electronics [10–12]. In addition, rapid changes in wireless communications and data transmission require radical adaptation of the materials used. The gradual development of 5G technology and the consequent transition to high-frequency bands require the development of materials with well-defined and adjustable dielectric properties along with flexibility and mechanical resistance [13,14]. In this context, low loss and high dielectric permittivity ($\varepsilon'$) are non-negotiable factors [13,15]. At present, materials

used for industrial microwave components include materials such as aluminum oxides (alumina), fused silicon, magnesium oxides, aluminum nitride, cordierite, and beryllium oxides, to name but a few [15,16]. The main asset of the above-mentioned materials is their high dielectric constant values in the GHz range [16]. However, despite their exquisite dielectric properties, serious flaws such as friction, complex and expensive manufacturing, low processing volume, and scaling limitations remain formidable obstacles to the broader use of these materials [15]. To address these problems, new, hybrid, organic–inorganic high-permittivity composites are employed [17,18]. In those materials, a high dielectric permittivity component is incorporated into a polymer matrix. Combining the well-known advantages of polymers such as flexibility, moldability, cost-effectiveness, low weight, and versatility, with the effective dielectric properties of an inorganic filler, these materials constitute an advantageous solution for large-scale production of affordable dielectrics [17,19]. To manipulate the general morphology of those composites, various methods of processing have been proposed, such as: (i) the addition of nanoparticles into polymers, where high-permittivity fillers uniformly disperse into the polymer matrix; (ii) the engineering of a filler–polymer interface by controlling the interaction between filler and polymer; the interface region affects the dielectric properties and is often considered the third stage of a composite material; and (iii) changing the surface of the coating by changing the substrate adsorption or polymer coating [20–22]. Thus, the inclusion of micrometer-sized particles into the polymer matrix was the chosen method of processing; mechanical mixing is a fast and easy way to combine polymers with dielectric oxides to produce flexible and highly dielectric materials.

However, regardless of the method chosen, the overall dielectric performance of the composite is mainly tuned by the filler percentage in the mixture. In addition, the vast majority of reports focus exclusively on the impact of strengthening agents on dielectric properties and the improvement of the preparation process in order to achieve the optimal dispersion of fillers in the matrix [23–30]. There are also a few studies investigating the dielectric behavior of composites under external stimuli; those studies focus primarily on electrical stimuli rather than mechanical stimuli [31]. However, the potential influence of mechanical stress on the dielectric properties is a challenging field, not fully investigated so far. To this end, it is necessary to perform combined experiments to measure the dielectric and mechanical properties simultaneously.

Given all the above, in this study, polymer–ceramic composites were produced from a simple mechanical mixture of the two components. In particular, we mixed commercially available silicone caulk, a widely used polymer with a low dielectric constant, with barium titanate (BTO) powder, a well-known ceramic with a high dielectric constant, which has been widely used in many studies to improve dielectric properties [14,32–45]. The mechanical mixing was performed manually until a fair homogeneity was achieved. Consequently, by simplifying the manufacturing process as much as possible, the possibility of using these composites on a large scale is provided. For the characterization of dielectric properties, several samples were generated, with varying quantities of BTO filler loading. It was found that the dielectric constant improved with increasing concentrations of BTO, reaching values which are comparable to other state-of-the-art dielectric materials [15,16]. For mechanical characterization of composites, samples of different filler percentages were subjected to tensile tests. It was found that BTO inclusions reduced the overall mechanical strength of composites. In contrast, combined dielectric permittivity and mechanical test experiments showed that the dielectric permittivity of composites remains almost unaffected by mechanical deformation, suggesting their ability to be used under tensile conditions. Therefore, the concept of flexible, easy-to-manufacture, and low-cost composite materials with improved dielectric properties that can potentially be used in the microwave components industry is presented here.

## 2. Materials and Methods

### 2.1. Sample Preparation

In order to prepare composite samples, a high-purity BTO powder with an average particle size of <2 m (purchased from Sigma-Aldrich, St. Louis, MO, USA) was used as a filler. In addition, commercial silicone (BISON silicone universal, Bolton adhesives, Rotterdam, the Netherlands) was used as a polymer matrix in appropriate quantities. After being weighed accurately, the two components were mechanically mixed (with a spatula) for approximately 15 min, until the mixture reached homogeneity. After that, the mixture was suffused using a spatula, such that parallilepidoid samples (typical dimensions 100 mm × 20 mm × 2 mm) were formatted. These samples were exposed to overnight drying at ambient temperature. The next day, the samples were placed in a drying oven at ~50 °C for 2 h to avoid any possibility of moisture being trapped in the samples. The above-mentioned procedure is shown in Figure 1.

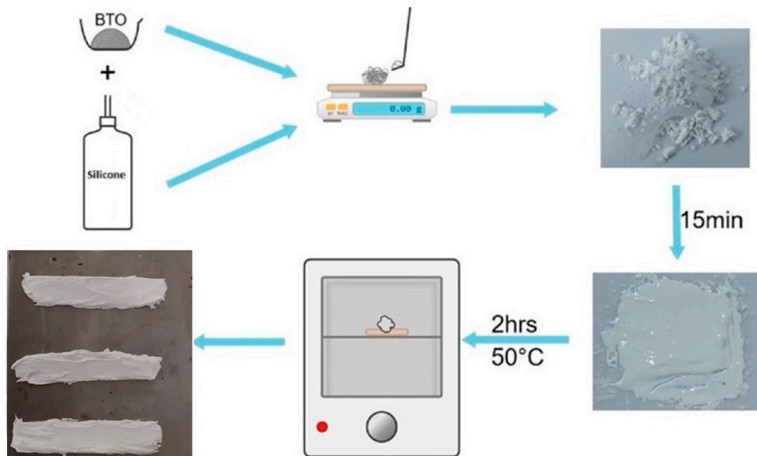

**Figure 1.** Schematic representation of fabrication process; the ingredients used to prepare BTO/silicone blends, the steps during mechanical mixing, and the final form of the constructed composites.

Following the above-described procedure, several samples, with varying BTO filler loading were produced ($m_{BTO}/m_{silicone}$) in a mixing range from 0 to 60/40. For each composite presented in the current study, three samples with almost identical dimensions and filler loading were produced. Here it must be noted that, for each filler loading, three different samples were produced. Apart from the samples, which were destroyed through mechanical stress experiments, all the other samples were subjected to dielectric permittivity measurements as well as combined experiments (as described in the following sections). Considering that each sample was measured three times, the overall error obtained is in the range of 5 to 10%, depending on the loading, suggesting a good reproducibility.

### 2.2. Characterization

#### 2.2.1. Scanning Electron Microscopy Measurements

The cross-sectional morphology of the samples was examined by scanning electron microscopy (SEM). To this end, a field emission scanning electron microscope (FE-SEM, JEOL JSM-7000F) equipped with an INCA microanalysis system (Oxford Instruments, Abingdon, UK) were used at several magnification scales.

#### 2.2.2. X-ray Diffraction Experiments

For the XRD measurements, a Bruker D8 Advance Diffractometer (Bruker Optik GmbH; Rosenheim, Germany) was used. The device was equipped with TWIN-TWIN technology and a Cu sealed tube source (wavelength 1.54 Å), operating at 40 kV and 35 mA. The powder configuration was used in theta-2theta mode in the range of 20–80 degrees for 2theta, with a step of 0.02 degrees and a speed of 3 s/step.

### 2.2.3. Raman Spectroscopy Experiments

Each BTO/silicone sample was placed on stainless-steel microscope slides and Raman measurements were performed at room temperature using a LabRAM HR Evolution Confocal Raman Microscope (LabRAM HR; HORIBA FRANCE SAS, Longjumeau, France) using a 532 nm laser source with a maximum power of ~30 mW and a $50\times$ microscopic objective lens (LMPlanFLN $50 \times 0.5$, Olympus) with a 0.5 numerical aperture (0.5 NA). The laser spot size, referring to the microscope resolution, was approximately 1.7 μm laterally and ~2 μm axially. A 600 groves/mm grating resulted in a Raman spectral resolution of around 2 cm$^{-1}$. The Raman spectral range was set from 50 to 3050 cm$^{-1}$, resulting in two optical windows per point. The acquisition time for each measurement was 5 s, with three accumulations at each point. The maximum power on the sample surface was 15 mW, 3 mW, 0.96 mW, or 0.3 mW, depending on the laser intensity applied (50%, 10%, 3.2%, or 1% respectively). Before Raman measurements, the wavelength scale was calibrated using a silicon standard (Silchem Handelsgesellschaft mbH, Freiberg, Germany) (520.7 cm$^{-1}$). Processing and analysis of the acquired raw Raman spectra were achieved through the instrument's original software (LabSpec, version LS6, Horiba, Lille, France). Initially, smoothing under a Gaussian filter with a kernel of five points (denoise at 5) was used, where cosmic rays were removed and the background was removed using a baseline correction at the sixth-order polynomial function. Finally, a shift to zero and a unit vector applied.

### 2.2.4. ATR Spectroscopy Experiments

In addition, ATR transmittance experiments were performed using a Bruker Vertex 70v FT-IR vacuum spectrometer (Bruker, Billerica, MA, USA), equipped with an A225/Q Platinum ATR unit with single reflection diamond crystal. Non-uniformly formed solid BTO/silicone samples were evaluated in total reflection mode, allowing for the infrared analysis of the samples. The spectral range was 4000–350 cm$^{-1}$, and the interferograms were collected at 4 cm$^{-1}$ resolution (8 scans). Before scanning the samples, a background measurement was recorded using a diamond crystal as a reference. For each measurement, the samples were carefully placed under the ATR press, while after every measurement, the sample area and the tip of the A225/Q ATR unit were cleaned with pure ethanol (Et-OH; Sigma-Aldrich, Munich, Germany).

### 2.2.5. Dielectric Constant Measurements

Dielectric constant measurements were performed, employing the open-ended coaxial probe method. An open-ended coaxial probe (N1501A dielectric probe kit, Keysight Technologies, Santa Rosa, CA, USA) was connected to a 2-port VNA (P9372A Streamline Vector Network Analyzer, Keysight Technologies, Santa Rosa, CA, USA). Dielectric permittivity values (real and imaginary part) were automatically calculated through the accompanying VNA software (N1500A Materials Measurement Software Suite, Keysight, Santa Rosa, CA, USA). VNA covered the frequency band between 1 MHz and 9 GHz, combining the so-called L, Ss and C bands of the electromagnetic spectrum, which are used for GPS, mobile phones (GSM), weather radar and microwave devices/communications, long-distance radio telecommunications such as satellite communications, Wi-Fi devices, cordless telephones, etc. During the whole process, the verticality of the sample probe was guaranteed; all measurements were taken at ambient temperature.

### 2.2.6. Mechanical Testing

The mechanical properties of the specimens were estimated through tensile tests by means of a universal testing machine (BKWW-5S, BIOBASE, Jinan, China). The testing machine was equipped with a 5 kN load cell, while the cross head speed was set to 20 mm/min. The corresponding stress–strain curves for samples with different BTO loads can be found in Section 3.

### 2.2.7. Combining Experiments

In order to investigate the impact of mechanical stress on dielectric properties, a combination of dielectric permittivity measurements under tensile stress was carried out. Two different methods were employed: (a) the open-ended coaxial probe method in the frequency range from 1 MHz to 9 GHz; in this case, the sample was placed in the universal testing machine while a constant tensile force was applied, while the dielectric constant was measured simultaneously for every ~0.5 cm of additional elongation (Figure 2a). It should be noted that it was important to reposition the probe at the same point on the sample surface every time the specimen was extended further. (b) The parallel-plate capacitor method, using a precision LCR meter (TH2829C, Changzhou Tonghui Electronic, Changzhou, China) at 1 MHz; each sample was placed in a universal testing machine, and two conductive plates (operating plate area: $A$ = 13.3 mm × 11.4 mm) were attached, one on each side of the sample, forming a plate capacitor (Figure 2b). The capacitor plates were connected to the LCR bridge with BNC cables. An AC signal of known amplitude and frequency was applied to the capacitor, allowing for both capacitance $C$ and dissipation factor $D$ to be recorded. Then, the real part of the dielectric constant, $\varepsilon\prime$, could be extracted, through the relation $\varepsilon' = C \cdot l / \varepsilon_o \cdot A$, where $A$ is the operating area of the capacitor plates; $l$ is the distance between the plates, which equals the sample thickness; and $\varepsilon_o$ is the dielectric constant of the vacuum. Moreover, the imaginary part of the dielectric permittivity was calculated through the relation $\varepsilon'' = D \cdot \varepsilon\prime$.

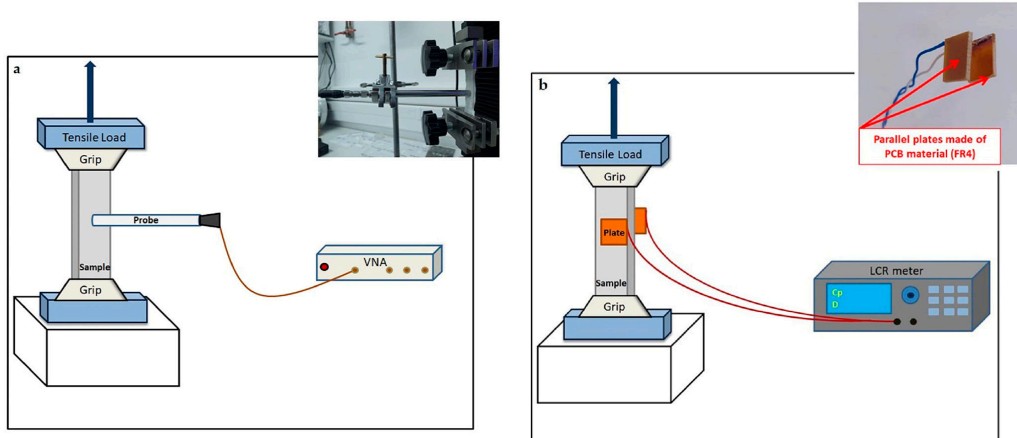

**Figure 2.** Graphical drawing of the setup used for combined dielectric measurements employing (**a**) the open-ended coaxial probe method and (**b**) the conventional parallel plate capacitor method.

The parallel plate capacitor method is a simple, well-known, and accurate method for the determination of the dielectric constant at low frequencies. Therefore, the above-mentioned advantages of the method were exploited to check and verify that the dielectric constant measurements obtained from the combined experiments were accurate and trustworthy, and then we adapted it to the high-frequency range of interest. In both cases, the reliability of the results obtained from combining the experiments was ensured by the fact that the above setup is basically a superposition of two well-calibrated individual arrangements (open-ended coaxial probe/LCR meter and universal testing machine), the accuracy of which has been tested separately.

Finally, dielectric constant measurements were carried out under the influence of repetitive tensile loading, employing the open-ended coaxial probe method. The samples under investigation were placed in the universal testing machine, and by repeated elongations between two fixed points on the surface of the sample, the dielectric constant was measured every 25 cycles for a total of 100 cycles.

## 3. Results and Discussion

### 3.1. Characterization

Figure 3a depicts typical SEM micrographs of the cross-sectional morphology of the BTO/silicone = 70/30 sample. Air voids can be clearly seen, and they may be attributed to air bubble entrapment during mechanical mixing. The presence of such voids definitely leads to increased porosity of the sample. At higher magnification (i.e., Figure 3b), the sample shows a granular profile consisting of BTO clusters of various sizes. Nevertheless, the high porosity of the bulk BTO powder (Figure 3c) and the presence of grain clusters depicted in the corresponding figure inset lead to the presumption that the fine BTO agglomeration tendency could be attributed to this granular result. It should also be noted that zero pressure was applied throughout the mechanical mixing process, preventing these clusters from becoming byproducts of the overall preparation process. Moreover, the inset of Figure 3b shows the encapsulation of BTO grains, indicating the presence of silicone in the mixture. Similar SEM micrographs were obtained for all examined samples studied, regardless of BTO concentration.

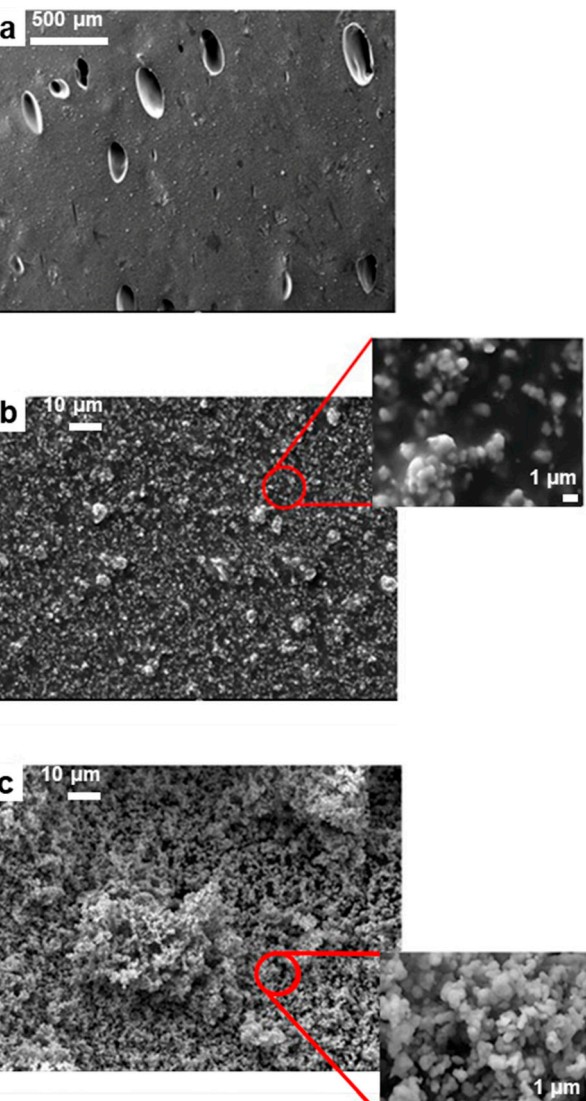

**Figure 3.** (**a**,**b**) Cross-sectional SEM micrographs of BTO/silicone sample at 70% filler content, at resolutions of 500 μm, 10 μm, and 1 μm (inset). The encapsulation of BTO in the polymer matrix can be seen in the inset. (**c**) SEM micrograph for bulk BTO powder at 10 μm and 1 μm resolution (inset).

Moreover, X-ray diffraction (XRD) patterns (Supplementary Figure S1) show that all marked diffraction peaks and their corresponding Miller indices were assigned to the BTO's perovskite structure [46,47]. Silicone does not show crystalline phases [48]. No other phases were observed, indicating that the mechanical mixture of silicone and BTO did not produce any chemical reactions, resulting in secondary crystalline products.

In addition, typical Raman spectra obtained for all BTO/silicone samples (Supplementary Figure S2a). The characteristic peaks and the corresponding vibrational modes are consistent with the literature for BTO and silicone respectively [49–52]. Furthermore, FTIR spectra were obtained for all samples studied (Supplementary Figure S2b). The characteristic peaks and the corresponding vibrational modes were in agreement with the literature for BTO and silicone respectively [47,52–56].

### 3.2. Dielectric Constant Measurements

Figure 4 shows the dielectric properties of BTO/silicone samples. In particular, the real part of the dielectric permittivity as a function of frequency is shown in Figure 4a. It is clear that there was an improvement in the dielectric constant with increasing BTO content. In particular, the highest permittivity values over the whole spectral range were observed for samples with filler loads of 60%. The increase in permittivity values due to BTO inclusions can be attributed to the increase in dipole content and therefore the overall polarization of the sample [32,38]. This behavior is consistent with previous reports on other BTO/polymer composites in the GHz range [33–37,42–45]. Furthermore, the BTO/silicone composites exhibited dielectric constant values comparable to those of other materials used for microwave applications, such as alumina [16].

In all cases, strong frequency dependence is observed when frequency increases, indicating a slight decrease in the dielectric constant value. It is known that the basic mechanism responsible for the electrical polarization of a dielectric material in the GHz regime is based on the interaction between electric dipoles. In this study, we focused on a relatively narrow frequency region (1–9 GHz), so the behavior of the dielectric constant was expected to exhibit moderate fluctuations, especially towards the upper frequency limit where the likelihood of being on the verge of a polarization mechanism transition was higher. Likewise, we observed a similar behavior for dielectric losses overfrequency (Figure 4c), especially for high BTO loadings. Moreover, as shown in SEM micrographs (Figure 3), the average particle size of the BTO powder used was <2 μm, which means that the grain size ranged from a few nm to 2 μm without any guarantee as to its homogeneity. The grain size of BTO had an important impact on dielectric properties, with the optimal grain size ranging between 0.8 and 1 μm [57,58]. It should be noted that the samples were not subjected to any kind of post-processing that could further improve their dielectric properties.

Figure 4b shows the dependence of the dielectric constant as a function of BTO/silicone mass percentage. The dielectric constant followed a gradual trend of rising in terms of BTO concentration, reaching a value of $\varepsilon'{\sim}8$ for highest BTO concentration. These results are in agreement with similar studies of BTO/polymer composites, which may have been produced by more complex methods [33–37,42–45], but are also comparable to other popular dielectric materials, such as alumina, borosilicate glass, steatite, cordierite, forsterite, mullite, and so on [15,59–62]. The minimum $\varepsilon'$ values are noted for the minimum filler content and vice versa, revealing a monotonous dependence on BTO load. The samples' dissipation factor is shown in Figure 4c. Loss tangent values tended to decrease alongside BTO concentration in the mixture; further reduction in conductivity due to the addition of BTO tended to reduce losses. Nevertheless, no general trend of losses in terms of filler content was observed. Dielectric loss behavior was consistent with similar studies reported in the literature [32,38].

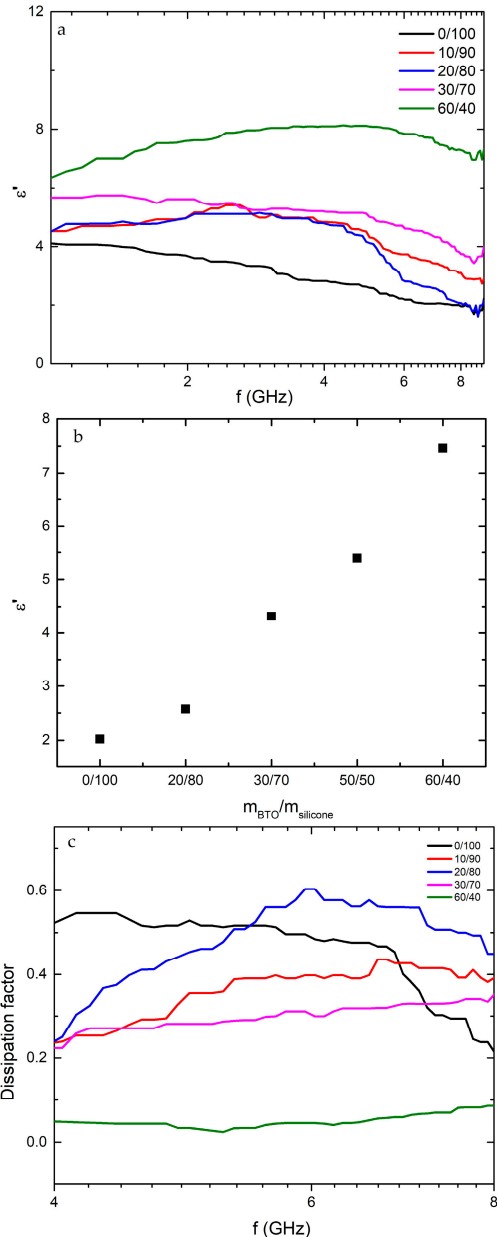

**Figure 4.** (**a**) Frequency dependence of the real part of dielectric permittivity $\varepsilon'$ for BTO/silicone samples (**b**) real part of dielectric permittivity $\varepsilon'$ over mass ratio $m_{BTO}/m_{silicone}$, as extracted from (**a**) at 7 GHz. (**c**) Dissipation factor for BTO/silicone samples. Loss tangent values for frequencies <4 GHz and >8 GHz were excluded due to high levels of noise in these spectral regions.

### 3.3. Mechanical Properties

Figure 5a shows the characteristic stress–strain curves for BTO/silicone samples with different mass ratios. It is clear that BTO inclusions led to suppression of the mechanical strength of the samples. The mechanical strength decreased with increasing BTO loading, which resulted in a consequent reduction in the breaking point, as shown in Figure 5b. (The maximum breaking point corresponds to 10% BTO; however, this specific point could be attributed to a measurement error.)

Increase of the filler content above 60% made the composite highly brittle and inappropriate for mechanical testing. Finally, the elongation during break reduction and the total stiffness of the samples caused by the addition of fillers was indicated in previous reports [31,63,64].

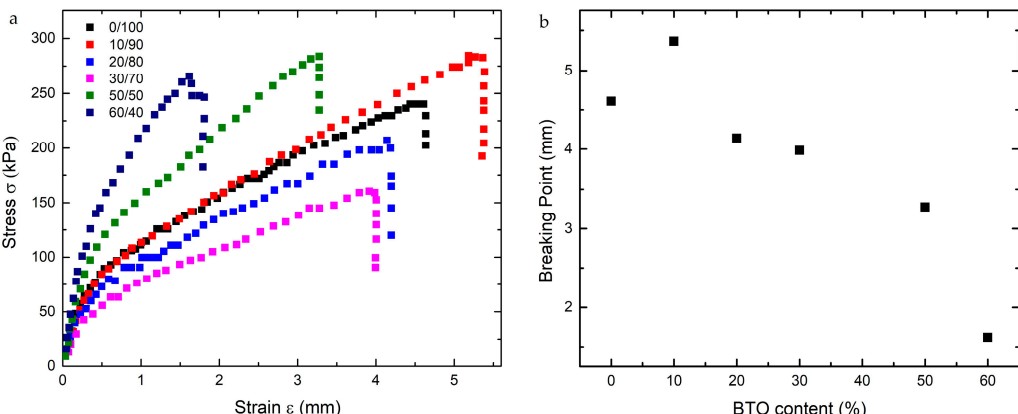

**Figure 5.** (**a**) Stress–strain curves for BTO/silicone samples for different mass ratios $m_{BTO}/m_{silicone}$; (**b**) breaking point of BTO/silicone samples for different BTO content.

Figure 6a shows the dielectric permittivity $\varepsilon'$ for various BTO/silicone samples during a tensile test at high frequencies (i.e., f = 3 GHz) as well as in the low-frequency regime (i.e., f = 1 MHz, Figure 6b). In both cases, a weak correlation between the dielectric and mechanical properties was observed. The values of the dielectric constant seem to have been relatively stable and independent of mechanical deformation, consistent with other studies for BTO/silicone composites [25,65]. In general, the correlation between dielectric and mechanical properties in a composite is challenging to observe experimentally, and even more so to explain. In this study, the BTO particles (exhibiting high dielectric constant ~100 [66]) were dispersed into the silicon rubber matrix (with a dielectric constant of ~2 [67]). The overall dielectric constant of the composites lays in the range 2–12, even for filler loadings as much as 60%. Therefore, it seems that the dielectric constant of the composite is mainly affected by the dielectric behavior of the matrix, even for high filler loadings. Rough estimations of the composites' dielectric constant (not shown here), using appropriate mixing models [33,39,68], corroborate the experimental results, enhancing the above scenario. On the other hand, experiments regarding the dielectric constant of pure silicone with respect to the sample elongation show that the dielectric constant is nearly unaffected by the sample stretching, suggesting a weak correlation between dielectric and mechanical properties (Supplementary Figure S3). Therefore, it is most likely that the dielectric constant of the BTO/silicone composites would follow the same trend. Furthermore, $\varepsilon'$ values were proportional to the BTO load, supporting the results previously mentioned (Figure 4b). The maximum value was obtained for the maximum filler content (60% BTO), while a slight change equal to an extension of 25 mm was observed regarrding both frequencies. This behavior can be attributed to the presence of polymer in the mixture; similar dielectric reactions for mechanical load-free silicone rubber have been reported in the literature [69]. Therefore, an improvement to dielectric constant is obtained without a significant change to mechanical behavior.

Figure 7 shows the real part of the dielectric permittivity in relation to the mass ratio of BTO/silicone samples exposed to a cyclic tensile load for a total amount of N = 100 cycles. It is clear that repeated stretching of the sample had no direct effect on the dielectric constant compared to the tension-free $\varepsilon'$ values. Such behavior is reported in the literature for similar BTO/silicone composites under mechanical deformations [25,65]. In addition, the dielectric constant values increased with increasing BTO loading, corroborating the observations previously shown (Figure 4a,b). The presence of slight differences in $\varepsilon'$ values at 25, 50, 75, and 100 cycles for a fixed mass ratio could be attributed to experimental configuration restrictions. It should also be noted that although fatigue failures can occur even in the elastic region of a material, this was not the case here. The flexibility of the polymer matrix provided non-destructive mechanical deformation, so samples could withstand more than 100 cycles of cyclic loading without adverse consequences on the dielectric properties.

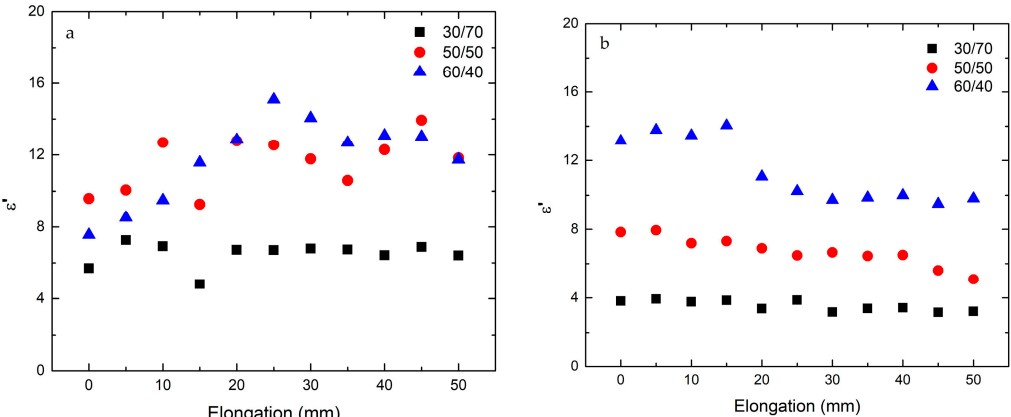

**Figure 6.** Real part of dielectric permittivity as a function of uniaxial elongation of the samples at different mass ratios ($m_{BTO}/m_{silicone}$) for a fixed frequency value of 3 GHz (**a**) and 1 Mhz (**b**), as measured via the open-ended coaxial probe and the parallel plate capacitor method, respectively.

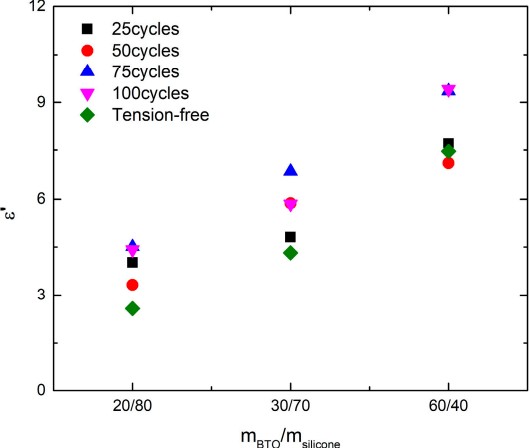

**Figure 7.** Dielectric constant values for BTO/silicone samples of different mass ratios ($m_{BTO}/m_{silicone}$ = 20/80, 30/70, 60/40) after 100 repetitions of cyclic stress at a fixed frequency value of 8.5 GHz.

## 4. Summary and Conclusions

In this work, mechanical mixing was used to produce BTO/silicone composites. To investigate the effects of BTO additions on dielectric and mechanical properties, samples with different BTO/silicone mass ratios were produced. The samples were oven-dried overnight without any further post-processing. The use of molds was not considered necessary in this study.

The cross-sectional morphology of the composites was characterized by SEM experiments. The corresponding SEM micrographs show the appropriate dispersion of BTO inclusions in the polymer matrix, which confirms the efficiency of the preparation process in terms of homogeneity. However, an increase in the porosity of the samples due to air bubble entrapments was also observed. The crystal structure of the composite was examined by XRD experiments. The crystallinity of BTO remained unchanged during treatment, as indicated by derived XRD patterns. On the other hand, silicone made no contribution to the obtained diffraction peaks. The molecular structure of the samples was studied using Raman and FTIR spectroscopy. The resulting characteristic peaks and corresponding modes were identified in relation to similar BTO/silicone mixtures in the literature, confirming that the compounds were composed of two distinct non-interactive phases.

Dielectric and mechanical measurements were carried out for all samples at room temperature. Experimental results show that the real part of dielectric permittivity was

improved with an increase in filler content, which is close to values of other state-of-the-art dielectric materials. Dielectric constant values are found to be most frequently independent in most of the frequency band studied. The imaginary part of the dielectric constant of BTO/silicone composites was found to be higher than that of other materials commonly used in the fields of radio frequency and photonics, but does not make this deviation prohibitive [16]. It is necessary to further develop methods to minimize losses.

To study mechanical properties, uniaxial tensile tests were performed in the presence of BTO at different mass ratios. The experimental results showed a significant reduction in the breaking point, combined with an expansion of the elastic region of the samples in relation to the filling content.

Last but not least, the contribution of combining experiments cannot be overestimated in terms of reducing the gap between existing dielectric properties and mechanical load. The dielectric constant was combined with a measurement at ambient temperature. The results showed that the dielectric constant of BTO/silicone composites has little dependence on the applied mechanical deformations, with $\varepsilon'$ values being comparable to tension-free specimens. Therefore, the combination of experiments provides valuable information on the impact of potential changes in the microstructure of the material on dielectric behavior; in our case, it remained unchanged. Thus, the inclusion of BTO as a ceramic filler in a flexible polymer matrix enables the production of highly stretchable composite materials with enhanced dielectric properties, offering a cost-effective alternative for microwave components.

**Supplementary Materials:** The following supporting information can be downloaded at: https://www.mdpi.com/article/10.3390/cryst14020160/s1, Figure S1: XRD patterns for BTO/silicone composites at various mass ratios ($m_{BTO}/m_{silicone}$), Figure S2: (a) Raman spectra for BTO/silicone composites at various mass ratios. Green highlighted peaks are assigned to silicone. (b) FTIR spectra for BTO/silicone composites at various mass ratios. Green-labeled peaks correspond to silicone, Figure S3: Dielectric constant as a function of uniaxial elongation for pure silicone, at 3 GHz.

**Author Contributions:** Conceptualization and methodology Z.V. and G.K.; validation, A.D., Z.V., K.K. and A.M.; investigation, A.D., Z.V., K.K. and A.M.; resources, G.K.; data curation, A.D. and Z.V.; writing—original draft preparation, A.D.; writing—review and editing, A.D., Z.V., G.K. and K.K.; project administration, Z.V. and G.K.; funding acquisition, G.K. All authors have read and agreed to the published version of the manuscript.

**Funding:** This project was funded by the project "METAmaterial-based ENERGY autonomous systems (META-ENERGY) (Project ID 2936)" which is implemented under the 2nd Call for H.F.R.I. "Research Projects to Support Faculty Members & Researchers" funded by the Operational Programme "Competitiveness, Entrepreneurship and Innovation" (NSRF 2014–2020). Moreover, this work was supported by proposal number 101092339 (Exploit4InnoMat), under the call: HORIZON-CL4-2022-RESILIENCE-01 (topic: HORIZON-CL4-2022-RESILIENCE-01-20; type of action: HORIZON-IA).

**Data Availability Statement:** The raw data supporting the conclusions of this article will be made available by the authors on request.

**Conflicts of Interest:** The authors declare no conflicts of interest.

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
