# Peer review of "Dielectric Behavior of Stretchable Silicone Rubber–Barium Titanate Composites"

_crystals, doi:10.3390/cryst14020160_

Round 1

Reviewer 1 Report

Comments and Suggestions for Authors

In this paper, the author comprehensively investigated the dielectric and mechanical properties of BTO/silicone composites with different filler loadings. The correlation between the dielectric and mechanical properties is also examined by combining experiments. Although the paper has been well written, it could be further improved by answering the following questions.

1. The dielectric constants at various frequencies are shown in Figs. 6-9. How are these frequencies chosen and why?

2. On Page 11, the author states that a weak correlation between the dielectric and mechanical properties is observed. Detailed explanation about this statement is needed.

3. Why did the author use two methods to measure the dielectric constant in the combining experiment?

4. The scale bars in the SEM images are not readable.

Comments on the Quality of English Language

Minor editing of English language is required.

Reviewer 2 Report

Comments and Suggestions for Authors

Dielectric elastomers have excellent properties, but the need for high operating voltage limits their practical application. In this study, polymer-ceramic composites were fabricated by simple mechanical mixing of two components: silicone sealant and barium-titanium (BTO) powder with different concentrations for experiments. The reviewed paper (manuscript ID: crystals-2839018, titled: Dielectric behavior of stretchable Silicone rubber-Barium Titanate nanocomposites) presents an original the results of research of prepare stretchable silicone rubber-barium titanate nanocomposites. The authors studied the effect of adding BTO mass ratios on dielectric and mechanical properties of nanocomposite at room temperature. The used research methods are appropriate. The research material is sufficient.  Figures are necessary and clear. The selection of literature is correct and adequate. I have no objections against the work by essence, but:

1. In my opinion, please consider highlighting the novelty in the abstract. Please consider reviewing the abstract and highlight the novelty, major findings, and conclusions. Please use numbers or % terms to clearly shows us the results in your work.

2. In general, what is the reproducibility of those experiments?

Reviewer 3 Report

Comments and Suggestions for Authors

The manuscript investigated the dielectric behavior of stretchable Silicone rubber-barium titanate nanocomposites. There are some interesting observations in the combining experiments of dielectric and mechanical properties. However, there are already many reports about the dielectric and mechanical properties of barium titanate-polymer composites. Especially no specific technique was used to disperse the barium titanate powders, leading to inconsistent results. I suggest that discussions about the dielectric and mechanical properties of the samples prepared by mechanical mixing and well-dispersed mixing should be added. The comments are as below:

1.     There are still many English grammatical and usage mistakes. I suggest it should be revised thoroughly by a native English speaker.

2.     The “nanocomposites” in the title was used. However, the particle size of raw material, barium titanate, is below 2 μm which does not meet the nano-sized range.

3.     “Barium Titanate” and “Silicon” should be revised as “barium titanate” and “silicon”.

4.      Line 66: “coating film surface” is difficult to follow. I suggest it should be revised.

5.     Lines 77-80: “However, the inevitable impact of mechanical strain on the microstructure of the composites and, potentially, on the dielectric properties poses a highly intriguing question for investigation.” It suggests the change of the microstructure during applying mechanical stress should be investigated. The reason that the dielectric property remained stable under mechanical deformation (Fig.8) should be explained.

6.     Line 93: state of the art should be revised as state-of-the-art.

7.     The sample preparation and measurement of the dielectric properties at low frequencies (Fig. 8(b)) should be provided.

8.     I suggest Figs.4 and 5 should be removed to supplementary materials because the mechanical mixing process was used.

9.     The variation of the dielectric properties with the measuring frequencies in Fig.6 should be explained and discussed.

10.   Lines 341-344: “It is clear that BTO inclusions induced serious deterioration regarding the mechanical strength of the samples. The mechanical strength is inversely proportional to BTO loading in the mixture, resulting in a steep reduction of the breaking point as depicted in Figure 7b.” It is not consistent with Fig.7, which shows the maximum bending point occurred at 10% BT and the maximum yield point occurred at 30wt%.

Comments on the Quality of English Language

There are many English grammatical and usage mistakes. I suggest it should be revised thoroughly.

Round 2

Reviewer 1 Report

Comments and Suggestions for Authors

The author has addressed all my questions. I have no more comments.

Comments on the Quality of English Language

Minor editing of English language is required.

Reviewer 3 Report

Comments and Suggestions for Authors

The manuscript has been revised to meet the requirement of the comments. I suggest it can accepted.